# Limited sensitivity of somatosensory evoked potentials as disease monitoring biomarkers in hereditary spastic paraplegias

Fernando Augusto Marion Spengler[1,2,3], Samanta Ferraresi Brighente[3],
Ana Luiza Rodrigues Louzada[2], Maria Eduarda Ribeiro de Souza[3],
Isabela Possebon Bevilacqua[1,3], Diana Maria Cubillos-Arcila[1,3],
Jonas Alex Morales Saute[1,2,3,4,5]*

1 Graduate Program in Medicine, Medical Sciences, Universidade Federal do Rio Grande do Sul, Porto Alegre, Brazil, 2 Neurology Division, Hospital de Clínicas de Porto Alegre, Porto Alegre, Brazil, 3 Clinical Neurogenetics research group, Hospital de Clínicas de Porto Alegre, Porto Alegre, Brazil, 4 Medical Genetics Division, Hospital de Clínicas de Porto Alegre (HCPA), Porto Alegre, Brazil, 5 Department of Internal Medicine, Universidade Federal do Rio Grande do Sul, Porto Alegre, Brazil

* jsaute@hcpa.edu.br

## Abstract

### Introduction

Hereditary Spastic Paraplegias (HSP) are a group of genetic disorders leading to the degeneration of long motor and sensory tracts in a progressive course. Clinician-reported outcomes (ClinROs) are the most commonly used endpoints for monitoring these diseases, but they have low sensitivity to detect progression. Therefore, identifying new monitoring biomarkers with higher sensitivity to change is crucial. Our objective was to compare the progression of Somatosensory Evoked Potential (SSEP) latencies over time with ClinROs in HSP.

### Methods

A longitudinal study was conducted on 22 individuals with a genetic diagnosis (13 SPG4, 3 SPG5, 3 SPG7, 2 SPG10, and 1 cerebrotendinous xanthomatosis), with two evaluations over a 4-year interval of upper limb (UL) and lower limb (LL) SSEPs and the Spastic Paraplegia Rating Scale (SPRS) total score and motor items only (mSPRS).

### Results

In the follow-up time analysis, progression after 4 years was observed only for SPRS and mSPRS, with an annual progression of 1.12 points and 1.02 points, respectively. No statistically significant progression was observed for SSEPs. Disease progression modeled according to disease duration showed worsening in all outcomes. For each additional year of disease, the SPRS worsened by 0.834 points (95% CI 0.62 to 1.04, $p < 0.001$), mSPRS by 0.758 points (95% CI 0.55 to 0.96, $p < 0.001$), SSEP-UL

**Data availability statement:** All relevant data are within the manuscript and its Supporting Information files.

**Funding:** This study was funded by Fundo de Incentivo à Pesquisa e Eventos – Hospital de Clínicas de Porto Alegre (FIPE-HCPA; Grant Number: 2019–0081). Cubillos-Arcilla is supported by CNPq, and Bevilacqua IP by CAPES.

**Competing interests:** The authors have declared that no competing interests exist.

latency by 0.164 ms (95% CI 0.03 to 0.3, p < 0.001), and SSEP-LL latency by 1.343 ms (95% CI 0.74 to 1.93, p < 0.001). Results for the SPG4 subgroup were similar to those for the overall HSP group.

## Conclusion

The neurophysiological progression of sensory long tract dysfunction is even slower than the progression of motor findings measured by COAs in HSP. The low sensitivity to change of SSEPs identified suggests that they should not be used as primary endpoints in future clinical trials for disease-modifying drugs.

## Introduction

Hereditary Spastic Paraplegias (HSP) are a group of rare monogenic disorders with an estimated prevalence of two to five cases per 100,000 individuals [1]. These conditions are characterized by length-dependent degeneration of the corticospinal tracts and dorsal columns, resulting in spasticity and muscle weakness predominantly in the lower limbs, which may be accompanied by the involvement of other neurological systems in complicated forms [1,2]. Currently, there is no disease-modifying treatment, and the slow progression of the disease combined with the lack of clinical assessment outcomes and biomarkers with high sensitivity to change and defined clinical relevance represents a major challenge in designing and proving the efficacy of new therapies [1,3].

A recent systematic review highlighted the marked heterogeneity in the use of clinical outcome assessments (COAs) in HSP. Among these, the Spastic Paraplegia Rating Scale (SPRS), a clinician-reported outcome, was the most commonly used [4]. However, the low sensitivity of the SPRS to detect disease progression underscores the critical need to identify other COAs and biomarkers with greater sensitivity to change [2,5–8].

Evoked potentials may represent promising biomarkers, given their ability to evaluate the integrity of both motor and sensory tracts, which are primarily affected in HSP. Several studies have already investigated motor evoked potentials (MEPs) and somatosensory evoked potentials (SSEPs) in HSP [2,9–15]. In a recent cross-sectional study from our group, both MEP and SSEP were assessed, showing their ability to differentiate HSP patients from healthy controls. MEP latencies were altered in both upper and lower limbs, while SSEP latencies were only altered in the lower limbs.

The presence of a ceiling effect for lower limbs (LL) MEP, in which many patients showed absent potentials, combined with the strong correlation between SSEP-LL latencies and disease duration, suggested that SSEP could be the best candidate as a disease progression biomarker. Moreover, studies evaluating proprioceptive pathways in HSP have demonstrated significant correlations between somatosensory pathway dysfunction and motor disease severity [3], further supporting the relevance of somatosensory dysfunction assessment as a biomarker in these conditions.

Therefore, given the absence of longitudinal studies using evoked potentials in HSP, our objective was to evaluate the potential of SSEP as a disease monitoring biomarker in HSP through a prospective study with a four-year follow-up.

## Methods

A single-center, exploratory, longitudinal study was conducted using cases from the study by Brighente and collaborators [9]. The sample consisted of 22 individuals (from 13 families) with a genetic diagnosis of HSP. For 17 participants (12 SPG4, 3 SPG5, 1 SPG7, and 1 cerebrotendinous xanthomatosis), the first assessments were conducted from October 1, 2019, to February 1, 2021. The second assessments occurred from between October 1, 2023, and October 31, 2024, maintaining a 4-year interval with a ± 3-month window. For the remaining 5 individuals (1 SPG4, 2 SPG10, and 2 SPG7), a single evaluation was conducted between 2023 and 2024, and their data were included only in the analyses of progression according to disease duration. Data from a patient with SPG11 who participated in the previous study [9] were excluded from this work due to the patient's inability to return to the research center and because only MEP data were available from the baseline evaluation.

The study was approved by the Ethics in Research Committee of Hospital de Clínicas de Porto Alegre (CAAE:09999519.9.0000.5327), Porto Alegre, Brazil. All participants were verbally informed about the study conditions and signed a written consent form. In cases of participants under 18 years of age, consent was signed by their parents.

Given the exploratory design of the study, no single primary outcome was defined, and sample size estimations were not performed. Data regarding sex, age at the last examination, age at onset (first motor sign), disease duration, and history of peripheral neuropathy were collected from patients and relatives or retrieved from electronic medical records. Disease severity was assessed using the Spastic Paraplegia Rating Scale (SPRS, range: 0–52, increasing with severity) [16]. Additionally, the motor-SPRS (mSPRS) was analyzed, excluding items related to pain and sphincter control (range: 0–44).

### Electrophysiological procedures

SSEPs were recorded using the Neuropack M1 MEB-9200 system (Nihon Kohden, Japan). Electrical pulses of 0.2 ms were delivered at a frequency of 3 pulses per second, with intensities ranging from 2 to 20 mV, applied to the medial malleolus and wrists, targeting the posterior tibial and median nerves, respectively. An average of 200–250 potentials were performed and superimposed to ensure reproducibility of the stimuli. Central recording electrodes were placed on the scalp over the primary sensory cortex (Fz, Cz, C3, C4), with peripheral reference points at the Erb's point for the upper limbs (UL) and the popliteal fossa for the lower limbs (LL). The recording sensitivity was set at 2 µV and 5 ms per division, with filters for lower and higher frequencies set at 10–2500 Hz. Recordings were analyzed over a 100 ms timeframe. N20 peak latencies were used for the UL, and N45 peak latencies were used for the LL. All neurophysiological evaluations were performed by the same evaluator (ALL) to minimize measurement bias. In cases where SSEPs were absent, a ceiling latency value of 100 ms was imputed. If SSEPs were absent at baseline but present at follow-up (observed in one case), this was considered a technical error in the baseline evaluation, and the data were censored. The single point data from such cases were only used in the progression model according to disease duration.

### Statistical analysis

To evaluate disease progression using SPRS, SPRS-M, UL-PESS, and LL-PESS metrics, the difference between their values at the two assessments was calculated and divided by each patient's follow-up time. Normality of these progression rates was assessed using Q-Q plots and the Shapiro-Wilk test, and all were found to adhere to a normal distribution. Paired t-tests were applied to determine whether the mean changes in these metrics differed significantly from zero. Effect size measures of sensitivity to change were performed with the standardized response mean (SRM). Pearson's correlation coefficient was used to assess the relationship between the calculated progression rates.

To model the outcomes SPRS, SPRS-M, UL-PESS, and LL-PESS as functions of disease duration, Generalized Estimation Equations (GEE) models were fitted. The adequacy of these models was assessed using Simulated Envelope Plots for the residuals, and most of the models were considered satisfactory. For SPRS and UL-N20 outcomes, attempts to improve model fit involved applying arcsine and logarithmic transformations, respectively. These transformations improved adherence to a normal distribution but did not materially affect the conclusions, so both transformed and non-transformed data are presented. All analyses were repeated, considering only SPG4 patients. A significance level of 5% (95% confidence) was adopted.

Data analysis and graph generation were conducted using R software version 4.3.1, with the following packages: readxl (v1.4.3), tidyr (v1.3.0), dplyr (v1.1.2), plyr (v1.8.9), ggplot2 (v3.5.1), lme4 (v1.1.35.5), lmerTest (v3.1.3), nlme (v3.1.165), geepack (v1.3.12), lattice (v0.21.8), gridExtra (v2.3), gtsummary (v1.7.2), gt (v0.11.0), knitr (v1.48), gee (v4.13.27), glmtoolbox (v0.1.12), hnp (v1.2.6), scales (v1.3.0), ggplotify (v0.1.2), and plotly (v4.10.4).

## Results

The main demographic and clinical characteristics of the overall sample of patients and the subgroup patients with SPG4 are summarized in Table 1. The complete data, including genotype descriptions at the case-based level, are presented in S1 Table.

### Progression by follow-up time

In the analysis by follow-up time, statistically significant progressions were observed after 4 years of follow-up only for SPRS and mSPRS, with progression of 1.12 points and 1.02 points per year, respectively. No statistically significant progression was observed for the SSEPs (Table 2).

**Table 1. Clinical and Demographic Characteristics of the Patients in the Study.**

|  | Overall HSP | SPG4 |
|---|---|---|
| **Female Sex** | 10/22 (45.5%) | 6/13 (46.2%) |
| **Age** |  |  |
| Mean (SD) | 41.4 (19.6) | 35.9 (22.2) |
| Median (Q25-Q75) | 48.5 (23.7 - 58.2) | 39.0 (10.5 - 58.0) |
| **Age at Onset** |  |  |
| Mean (SD) | 24.7 (16.8) | 21.4 (18.8) |
| Median (Q25-Q75) | 30.0 (6.5 - 40.0) | 29.0 (2.5 - 36.5) |
| **Pure Form** | 16/22 (72.7%) | 13/13 (100%) |
| **Disease Duration** |  |  |
| Mean (SD) | 16.7 (9.2) | 14.4 (9.6) |
| Median (Q25-Q75) | 18.0 (7.7 - 25.2) | 10.0 (6.5 - 23.0) |
| **SPRS** |  |  |
| Mean (SD) | 17.1 (8.9) | 14.4 (9.5) |
| Median (Q25-Q75) | 15.0 (10.5 - 25.2) | 14.0 (7.5 - 26.5) |
| **mSPRS** |  |  |
| Mean (SD) | 15.3 (8.4) | 16.3 (10.5) |
| Median (Q25-Q75) | 13.5 (9.7 - 23.2) | 12.0 (5.5 - 22) |
| **Peripheral Neuropathy** | 2/22 (9%) | 0/13 (0%) |

Age, age at onset and disease duration are depicted in years. SPRS: Spastic Paraplegia Rating Scale; mSPRS: motor Spastic Paraplegia Rating Scale; SD, standard deviation; Q, quartile.

**Table 2. Annual Progression of COAs and PESS by Follow-up Time.**

| Variable | Mean (CI 95%) | SRM (CI 95%) | p-value |
|---|---|---|---|
| SPRS | 1.12 [0.63 to 1.61] | 1.18 [0.74 to 2.18] | < 0.001 |
| mSPRS | 1.02 [0.61 to 1.43] | 1.26 [0.85 to 2.12] | < 0.001 |
| SSEP-UL Latency (msec) | −0.03 [−0.60 to 0.54] | −0.03 [−0.56 to 0.48] | 0.910 |
| SSEP-LL Latency (msec) | 0.63 [−1.55 to 2.81] | 0.16 [−0.36 to 0.89] | 0.545 |

SPRS: Spastic Paraplegia Rating Scale; mSPRS: motor Spastic Paraplegia Rating Scale; SRM, standardized response mean; SSEP-UL upper limbs somatosensory evoked potential. SSEP-LL lower limbs somatosensory evoked potential.

When analyzing the progressions over time between the outcomes under study, a statistically significant correlation was observed only between the progressions of the SPRS and mSPRS (Rho = 0.84, p < 0.001, Fig 1A) and between the progressions of the SSEP-UL latencies with the SSEP-LL latencies (Rho = 0.54, p < 0.05, Fig 1F). No statistically significant correlations were found between the progressions of clinical evaluation outcomes and the progressions of the SSEP latencies (p > 0.05 for all comparisons, Fig 1B-1E).

The results for the progression of COAs and SSEPs over time (S2 Table), as well as the correlations between their progressions (S1A-S1F Fig) in the SPG4 subgroup, were similar to those in the overall HSP group.

### Disease progression modeled according to the disease duration

In the analysis of disease progression modeled according to disease duration, significant progressions were found for all the variables of interest.

The SPRS progressed by 0.834 points (95% CI 0.62 to 1.04, p < 0.001, Fig 2A) and the mSPRS progressed by 0.758 points (95% CI 0.55 to 0.96, p < 0.001, Fig 2B) for each additional year of disease duration in the overall HSP group. The latency of the SSEP-UL increased by 0.164 ms (95% CI 0.03 to 0.3, p = 0.018, Fig 2C) and that of the SSEP-LL increased by 1.343 ms (95% CI 0.74 to 1.93, p < 0.001, Fig 2D) for each additional year of disease duration in the overall HSP group. See Table 3 for detailed results of the models.

The results for the progression of the SPRS with the arcsine transformation and for the SSEP-UL with the logarithmic transformation were similar to the untransformed results shown above and can be seen in S3 Table and in Figs 2A and 2C (dashed gray lines).

The results of the subgroup analysis for patients with SPG4 only were similar (see S4 Table, S2A-S2F Fig).

## Discussion

This is the first study to longitudinally assess SSEPs in Hereditary Spastic Paraplegia (HSP). Although no progression in SSEP was detected over 4 years for both upper (UL) and lower limbs (LL), a very small annual progression was observed when disease duration was modeled. In contrast, both SPRS and mSPRS progressed when analyzed by follow-up time, as well as in the disease duration model, suggesting a greater sensitivity to change in COAs than in the studied neurophysiological biomarkers.

### Progression of HSP in clinical outcomes assessments

In this study, we found a progression of 1.12 points per year in SPRS in the follow-up time analysis and 0.83 points per year in the progression model according to disease duration. Several studies have evaluated SPRS progression in HSP, with similar results. In a recent study from our group, we found a progression of 0.43 points per year over 4.5 years in a sample predominantly composed of patients with SPG4 [9]. In a cohort of patients with childhood-onset, progression ranged from 1.08 per year in SPG4 to 1.37 in SPG11 [5]. In a study with 34 patients with SPG5, the SPRS progression

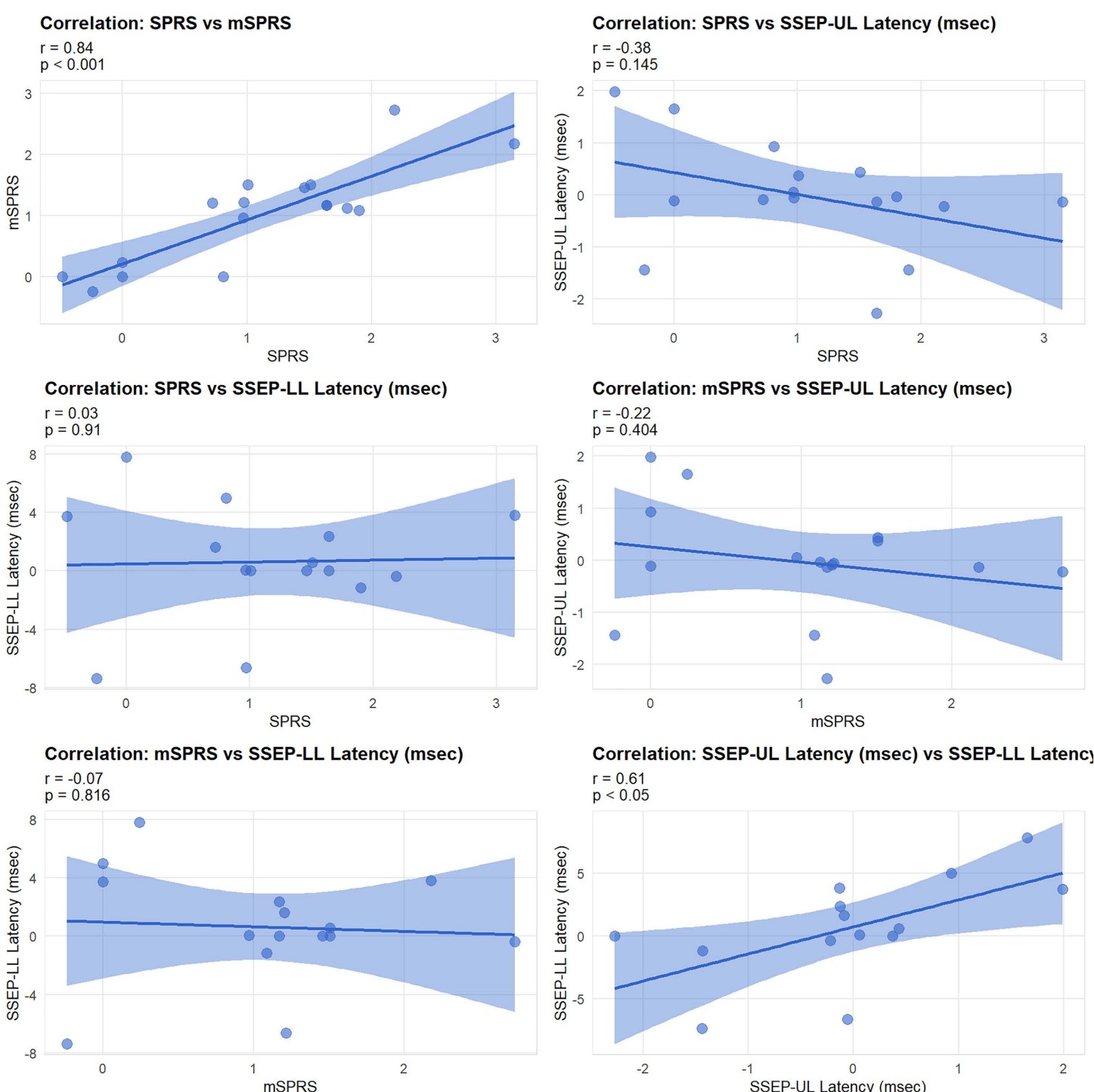

**Fig 1. Correlation of Progressions Between Variables Over Follow-up Time.** SPRS: Spastic Paraplegia Rating Scale; mSPRS: motor Spastic Paraplegia Rating Scale. SSEP-UL upper limbs somatosensory evoked potential. SSEP-LL lower limbs somatosensory evoked potential.

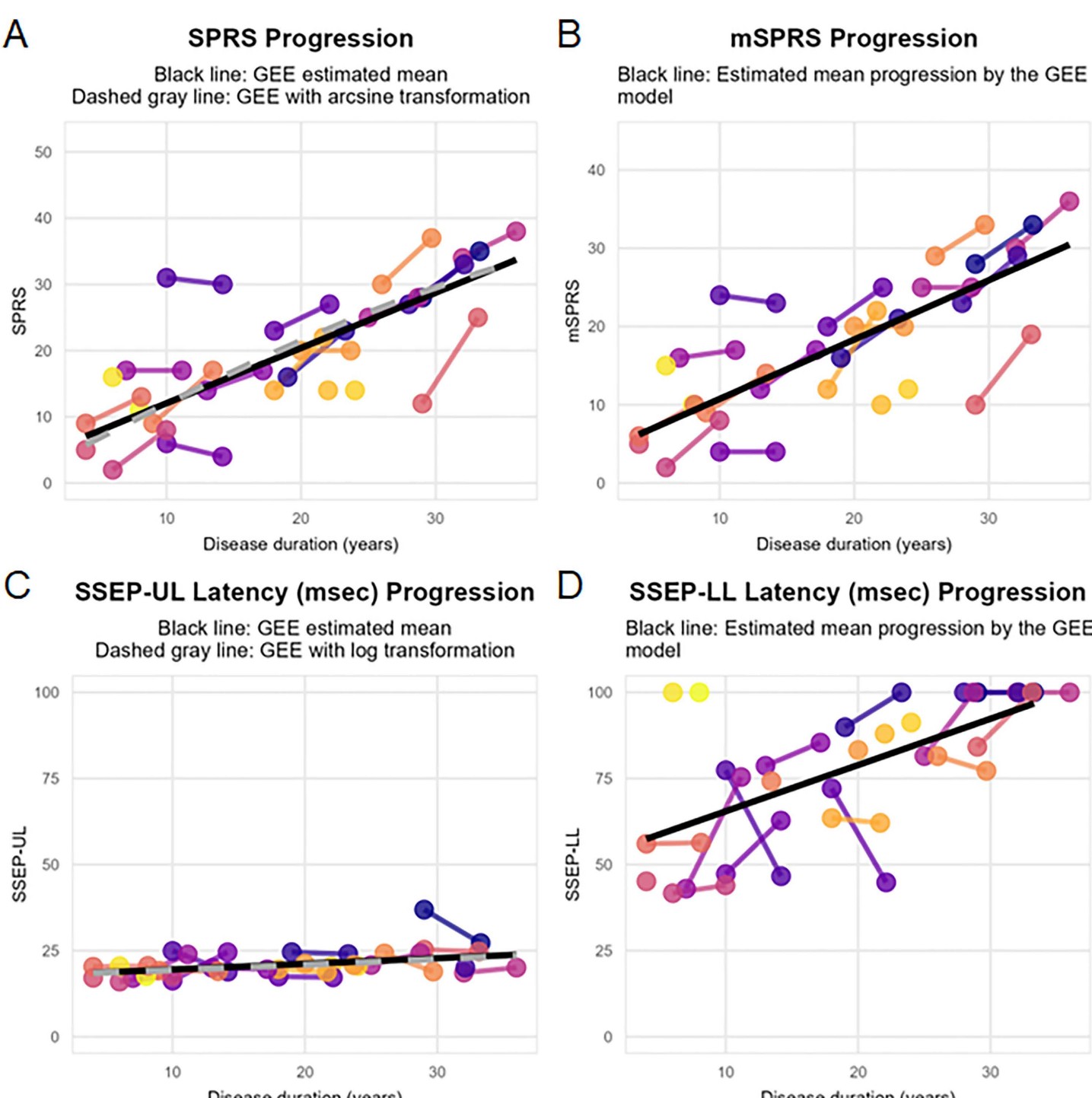

**Fig 2. Disease progression modeled according to disease duration.** Progression of the **A)** Spastic Paraplegia Rating Scale (SPRS), **B)** the motor-SPRS (mSPRS), **C)** the latency of the SSEP-Upper Limb (SSEP-UL) and **D)** the latency of the SSEP-Lower Limb (SSEP-LL) over time. Dashed gray lines represent the results after applying arcsine or logarithmic transformations, as shown in the S2 Table. GEE Generalized Estimating Equations.

**Table 3. Disease progression modeled according to the disease duration.**

| Variable | Estimate | Standard Error | Statistic | Mean (CI 95%) | p-value |
|---|---|---|---|---|---|
| SPRS | 0.834 | 0.1090 | 58.5676 | 0.62 to 1.04 | < 0.001 |
| mSPRS | 0.758 | 0.1064 | 50.8303 | 0.55 to 0.96 | < 0.001 |
| SSEP-UL Latency (msec) | 0.164 | 0.0694 | 5.602124 | 0.03 to 0.3 | 0.018 |
| SSEP-LL Latency (msec) | 1.343 | 0.3032 | 16.6281 | 0.75 to 1.93 | < 0.001 |

SPRS: Spastic Paraplegia Rating Scale; mSPRS: motor Spastic Paraplegia Rating Scale. SSEP-UL upper limbs somatosensory evoked potential. SSEP-LL lower limbs somatosensory evoked potential. GEE Generalized Estimating Equations.

was 0.56 points per year in a cross-sectional evaluation. In a subgroup of 21 patients with multiple assessments, the progression was 0.8 points per year. In a group of 30 patients with HSP (mainly complex forms and some without a molecular diagnosis), progression of 1.17 points per year was identified [7]. A cross-sectional evaluation correlating SPRS with disease duration found 0.83–1.7 points per year in a study with 278 SPG4 patients [2]. A subgroup with 28 patients who had 4 or more evaluations showed a progression of 0.5 points per year [2].

### SSEPs as biomarkers of HSP progression severity

Several studies have assessed the SSEP cross-sectionally in HSP, with varying results, primarily due to the heterogeneity of the studies [2,9–15]. A case-control study in 2007 compared 12 patients with SPG4–16 healthy controls. No statistical difference was found between the groups for the SSEP in the upper limbs (UL). In cortical recordings for the lower limbs (LL), 25% were absent. No statistically significant reduction in latencies was found, although the amplitude showed a significant decrease [10]. The largest published study to date with neurophysiological assessment of HSP included 357 patients, of whom 18.2% showed abnormalities on SSEP. However, the study did not perform a quantitative analysis of the SSEP, only a qualitative analysis, categorizing the results as abnormal or normal [2]. In a case series of 128 patients, the cortical latencies (N20) of median nerves were prolonged in 9% and absent in 9%, while the tibial nerve latencies (P40) were prolonged in 7% and absent in 29%, with a clinical correlation to sensory deficits [11]. Two studies assessed SSEP in SPG5, one with 3 patients found a delayed conduction in one, inconclusive in another, and normal in the third. The other study, with 4 patients, showed abnormalities in all; these patients had a longer mean disease duration (37.5 years) [13,14]. In an assessment with 6 patients with SPG11, all presented results within the normal range. It is important to note that the patients were young (mean age 23.3 years) and had a short disease duration (mean 4 years) [12]. A publication with 9 patients with SPG35 identified that 2/9 had abnormal SSEP, but the details of the findings were not provided [15]. Our group recently published the results of the cross-sectional analysis of data from this project, in which 18 HSP patients (12 SPG4) and 21 healthy controls were analyzed. SSEP in the upper (UL) and lower limbs (LL) was assessed in 17 of these individuals, and the data were correlated with disease duration and clinical scales (SPRS). Only the SSEP-LL presented significantly longer latencies compared to the control group, with a strong positive correlation between the latency of SSEP-LL and disease duration, but no correlation with COAs [9]. Despite the lack of correlation between SSEP and COAs related to motor severity, another recent study from our group showed that proprioceptive alterations impacting balance in HSP patients correlated with motor severity of HSP as assessed by COAs [3], suggesting that the degeneration of both motor and sensory pathways might occur at similar rates, reinforcing the idea of evaluating sensory pathways as disease monitoring biomarkers.

To the best of our knowledge, this is the first study to longitudinally assess the progression of SSEP findings in patients with HSP, and the obtained data indicate that a 4-year period was not sufficient to detect progression in latencies. Furthermore, there was no correlation between the progression of SPRS and mSPRS with the SSEPs. Significant progression in

SSEP latencies was only identified when progression was modeled according to disease duration, with small effect sizes. Taken together, these data suggest that the degeneration of long sensory pathways in HSP is progressive but occurs at a slower rate compared to the motor findings measured by COA SPRS and mSPRS.

### Study limitations

The major study limitation was its small sample size and exploratory design. The majority of our sample consisted of patients with SPG4 and moderate disability. Future studies evaluating other subtypes of HSP, particularly those with faster progression such as SPG11, are important to more comprehensively assess the role of SSEPs as a disease monitoring biomarker. Additionally, as 37% of patients had absent SSEP-LL results in the final assessment, it would be important for future studies to evaluate patients in the early stages of the disease or even pre-symptomatic individuals to avoid this ceiling effect. Since the ceiling effect in the baseline population [9] had been marked for the MEP, we chose not to assess it in the present study. However, we consider essential to evaluate MEP in futures studies if involving subjects in early or pre-symptomatic stages. Another limitation is that abnormalities in sensory nerve conduction studies might have affected SSEPs; however, since only 2/22 individuals in the study presented peripheral neuropathy, both with complex HSP, and the SSEPs results of the overall HSP and the SPG4 subjects (all with pure HSP and with no evidence of peripheral neuropathy) were similar, it is unlikely that the lack of correction for sensory NCS have influenced the study results in a significant manner. Of note, S1 Table provides comprehensive clinical data to enable broad and transparent access to case-based information, as well as to potential biases that may be influencing the findings.

### Conclusion

The progression of sensory tract dysfunction is slower than the progression of motor findings measured by COAs such as SPRS in HSPs, particularly in SPG4. This is an important insight into how neurodegenerative processes occur in the different pathways involved in this condition. The low sensitivity to change in SSEPs suggests that they should not be used as primary outcome biomarkers in future clinical trials for disease-modifying drugs in HSP.

### Supporting information

**S1 Fig. Correlation of Progressions Between Variables Over Follow-up Time in the SPG4 Subgroup.** SPRS: Spastic Paraplegia Rating Scale; mSPRS: motor Spastic Paraplegia Rating Scale. SSEP-UL upper limbs somatosensory evoked potential. SSEP-LL lower limbs somatosensory evoked potential.
(TIF)

**S2 Fig. Disease progression modeled according to the disease duration in the SPG4 subgroup.** Progression of the A) Spastic Paraplegia Rating Scale (SPRS), B) motor-SPRS (mSPRS), C) latency of the SSEP-Upper Limb (SSEP-UL) and D) latency of the SSEP-Lower Limb (SSEP-LL) over time. Dashed gray lines represent the results after applying arcsine or logarithmic transformations, as shown in the S2 Table. GEE Generalized Estimating Equations.
(TIF)

**S1 Table. Detailed clinical and genetic information with raw neurophysiological data.**
(XLSX)

**S2 Table. Progression by Follow-up Time in the SPG4 Subgroup.**
(DOCX)

**S3 Table. Disease progression modeled according to the disease duration with data transformation.**
(DOCX)

**S4 Table. Disease progression modeled according to the disease duration in the SPG4 subgroup.**
(DOCX)

**S1 Checklist. STROBE Statement.**
(DOCX)

## Acknowledgments

The authors thank the patients and their families who participated in this long-term, challenging cohort study. The English language revision was conducted with the assistance of an AI language model developed by OpenAI.

## Author contributions

**Conceptualization:** Fernando Augusto Marion Spengler, Isabela Possebon Bevilacqua, Jonas Alex Morales Saute.

**Data curation:** Jonas Alex Morales Saute.

**Formal analysis:** Fernando Augusto Marion Spengler, Jonas Alex Morales Saute.

**Funding acquisition:** Jonas Alex Morales Saute.

**Investigation:** Fernando Augusto Marion Spengler, Samanta Ferraresi Brighente, Ana Luiza Rodrigues Louzada, Maria Eduarda Ribeiro de Souza, Diana Maria Cubillos-Arcila, Jonas Alex Morales Saute.

**Methodology:** Fernando Augusto Marion Spengler, Samanta Ferraresi Brighente, Jonas Alex Morales Saute.

**Project administration:** Jonas Alex Morales Saute.

**Resources:** Jonas Alex Morales Saute.

**Supervision:** Jonas Alex Morales Saute.

**Writing – original draft:** Fernando Augusto Marion Spengler.

**Writing – review & editing:** Samanta Ferraresi Brighente, Jonas Alex Morales Saute.

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
