## [Decision Letter · Decision Letter 0]

9 May 2025

Dear Dr. Saute,

Thank you for submitting your manuscript to PLOS ONE. After careful consideration, we feel that it has merit but does not fully meet PLOS ONE’s publication criteria as it currently stands. Therefore, we invite you to submit a revised version of the manuscript that addresses the points raised during the review process.

Your manuscript has been reviewed by two expert referees who raised several concerns, including some significant issues that need to be addressed. We ask that you carefully and thoroughly respond to each of the reviewers' comments in your revised submission.

As you prepare your revision, please ensure that your responses are clearly reflected in the manuscript itself. This will not only help readers benefit from the additional insights and clarifications but also demonstrate how the manuscript has been strengthened in response to the reviewers' feedback.

We look forward to receiving your revised manuscript.

Kind regards,

Jian Jing, Ph.D.

Academic Editor

PLOS ONE

Journal Requirements:

“This study was funded by Fundo de Incentivo à Pesquisa e Eventos-Hospital de Clínicas de Porto Alegre (FIPE-HCPA) – Grant Number: 2019–0081). Cubillos-Arcilla is supported by CNPq, and Bevilacqua IP by CAPES.”

4. In this instance it seems there may be acceptable restrictions in place that prevent the public sharing of your minimal data. However, in line with our goal of ensuring long-term data availability to all interested researchers, PLOS’ Data Policy states that authors cannot be the sole named individuals responsible for ensuring data access (http://journals.plos.org/plosone/s/data-availability#loc-acceptable-data-sharing-methods).

Reviewers' comments:

Reviewer's Responses to Questions

**Comments to the Author**

1. Is the manuscript technically sound, and do the data support the conclusions?

Reviewer #1: Yes

Reviewer #2: No

2. Has the statistical analysis been performed appropriately and rigorously?

Reviewer #1: Yes

Reviewer #2: No

3. Have the authors made all data underlying the findings in their manuscript fully available?

Reviewer #1: No

Reviewer #2: Yes

4. Is the manuscript presented in an intelligible fashion and written in standard English?

Reviewer #1: Yes

Reviewer #2: Yes

Reviewer #1: This is a well written study regarding the utility of SSEPs as potential biomarkers for HSPs. As with most HSP biomarker studies, sample size and heterogeneity are limitations due to the rarity and variability of the condition. The longitudinal aspect of the study is unique and helpful to demonstrate the inability of SSEPs to measure disease progression over time. A few minor comments below:

1. The presence of a possible ceiling effect for SSEPs may have impacted its ability to measure disease progression. Was there a correlation between SPRS scores and patients with absent SSEPs? (i.e. did patients with absent SSEPs have higher SPRS or mSPRS scores?)

2. The authors mentioned the presence of peripheral neuropathy in 3 patients (Study limitations line 300-301). This is relevant information to include in the Results section under Clinical characteristics. Similarly, number of patients with pure vs complex HSP is relevant too.

3. The authors report a statistically significant correlation between progression of SPRS and mSPRS (Results section, line 192). This is to be expected as the mSPRS is a subset of the SPRS. Was there a correlation between the mSPRS score and non-motor items in SPRS?

4. Was there correlation between SSEPs and non-motor items in SPRS?

5. In the discussion, the authors report their previous study finding of proprioceptive impairment correlating with SPRS scores (line 274-5). Was there overlap between the study populations? If so, was there correlation between proprioceptive impairment and SSEPs?

Reviewer #2: In this manuscript the author, Drs Fernando Augusto Marion Spengler et al., longitudinally assessed SSEPs in Hereditary Spastic Paraplegia.

The major study limitation was its small sample size. Moreover, the manuscript requires detailed genetic data which is currently non-existent. Finally, several references should be updated.

**Do you want your identity to be public for this peer review?** For information about this choice, including consent withdrawal, please see our Privacy Policy

Reviewer #1: **Yes: ** Sue Faye Siow

Reviewer #2: No

---

## [Author Response · Author response to Decision Letter 1]

4 Sep 2025

Response to Reviewers

Reviewer #1: This is a well written study regarding the utility of SSEPs as potential biomarkers for HSPs. As with most HSP biomarker studies, sample size and heterogeneity are limitations due to the rarity and variability of the condition. The longitudinal aspect of the study is unique and helpful to demonstrate the inability of SSEPs to measure disease progression over time.

Response: We thank the reviewer for the thoughtful comments and fully agree that these are the main strengths and limitations of our study.

A few minor comments below:

1. The presence of a possible ceiling effect for SSEPs may have impacted its ability to measure disease progression. Was there a correlation between SPRS scores and patients with absent SSEPs? (i.e. did patients with absent SSEPs have higher SPRS or mSPRS scores?)

Response: We thank the reviewer for the comment. In our previous study presenting cross-sectional SSEP data in HSPs (Brighente et al., 2021, PMID: 34847171), we reported a difference in lower limb SSEP (LL-SSEP) between cases and controls, as well as a strong and direct correlation with disease duration. A direct correlation between LL-SSEP and SPRS was also observed, but it did not reach statistical significance (Rho = 0.483, p = 0.068). At baseline in that study, only one patient showed absence of LL-SSEP. In the current study, 37% of the cases had absent LL-SSEP after four years of follow-up.

To address the reviewer’s comment, we calculated the mean SPRS score for patients with absent potentials at any evaluation, which was 25.58. In contrast, the mean SPRS score for patients with measurable potentials was 17.13. This difference was statistically significant (p = 0.015).

We agree with the reviewer that a ceiling effect may have influenced the results, and the data above support the interpretation that more severely affected patients were those who exhibited this ceiling effect. However, we believe this represents a methodological limitation—namely, that a significant proportion of patients may eventually show absent potentials over the course of the disease, which limits the utility of this tool as a biomarker of disease progression.

As this issue was already addressed in the limitations paragraph—where we suggest further studies in patients at earlier stages of the disease, when the ceiling effect is less prominent (as observed at baseline), and where LL-SSEP might eventually capture longitudinal worsening—we opted not to modify the main content of the manuscript. However, we have added Supplementary Table 1, which provides individual-level data, allowing interested readers to explore the raw data and perform additional analyses such as the one suggested.

2. The authors mentioned the presence of peripheral neuropathy in 3 patients (Study limitations line 300-301). This is relevant information to include in the Results section under Clinical characteristics. Similarly, number of patients with pure vs complex HSP is relevant too.

Response: We thank the reviewer for the comment. We agree with the observations and have added the corresponding information to Table 1 and Supplementary Table 1, which contains the complete data for all cases. We have also corrected the statement: only two cases had peripheral neuropathy.

3. The authors report a statistically significant correlation between progression of SPRS and mSPRS (Results section, line 192). This is to be expected as the mSPRS is a subset of the SPRS. Was there a correlation between the mSPRS score and non-motor items in SPRS?

Response: We thank the reviewer for the comment. We agree that the correlation was somewhat expected; nevertheless, we chose to retain it in the manuscript. Although the reviewer’s question regarding the correlation between the mSPRS score and the non-motor items in the SPRS is relevant and interesting, addressing these aspects would go beyond the scope of the current manuscript. The SPRS includes two patient-reported outcome (PRO) items related to bladder function and pain, which, in the original validation study of the scale (Schüle et al., 2006), were grouped into a distinct factor, separate from the other 10 items, as well as from the contracture item, which formed a third component. Our intention was to maintain consistency with the analytical approach used in our baseline report, and to avoid potential confounders in the total SPRS score, particularly regarding items 12 and 13. These two items are not only distinct in the original factor analysis of the scale but also may be influenced by symptomatic treatments that could affect their longitudinal scoring. Nonetheless, we performed a non-parametric correlation analysis between the mSPRS and items 12 and 13 across all timepoints assessed in the study. A weak direct correlation was observed (Rho = 0.335, p = 0.034). However, we did not modify the manuscript based on this comment.

4. Was there correlation between SSEPs and non-motor items in SPRS?

Response: There was no significant correlation (Rho = 0.198, p = 0.254, considering all timepoints analyzed in the study). We chose not to include these data in the manuscript for the reasons outlined in response to Comment 3.

5. In the discussion, the authors report their previous study finding of proprioceptive impairment correlating with SPRS scores (line 274-5). Was there overlap between the study populations? If so, was there correlation between proprioceptive impairment and SSEPs?

Response: There was sample overlap; however, the evaluation timepoints did not coincide, making this analysis unfeasible. The findings from the 3-year follow-up study (in contrast to the current 4-year study) on proprioceptive changes assessed via force platform showed deterioration in the parameters—unlike what we observed with LL-SSEP over 4 years. However, as those data are part of a different manuscript, have not yet been published, and do not match the timepoints analyzed in the present study, we are unable to include this information in the current publication.

Reviewer #2: In this manuscript the author, Drs Fernando Augusto Marion Spengler et al., longitudinally assessed SSEPs in Hereditary Spastic Paraplegia.

The major study limitation was its small sample size. Moreover, the manuscript requires detailed genetic data which is currently non-existent. Finally, several references should be updated.

Response: We thank the reviewer for the comment. As suggested by Reviewer 1, we have included Supplementary Table 1, which presents the complete dataset for the study population, and we have added the patients' genotypes along with the respective variant classifications based on the 2015 ACMG/AMP guidelines. In addition, we carefully reviewed the manuscript once more and updated some of the references in the Introduction, as suggested.

---

## [Decision Letter · Decision Letter 1]

8 Oct 2025

LIMITED SENSITIVITY OF SOMATOSENSORY EVOKED POTENTIALS AS DISEASE MONITORING BIOMARKERS IN HEREDITARY SPASTIC PARAPLEGIAS

PONE-D-25-06200R1

Dear Dr. Saute,

We’re pleased to inform you that your manuscript has been judged scientifically suitable for publication and will be formally accepted for publication once it meets all outstanding technical requirements.

Kind regards,

Jian Jing, Ph.D.

Academic Editor

PLOS ONE

Additional Editor Comments (optional):

Reviewers' comments:

Reviewer's Responses to Questions

**Comments to the Author**

Reviewer #1: All comments have been addressed

Reviewer #2: All comments have been addressed

2. Is the manuscript technically sound, and do the data support the conclusions?

Reviewer #1: Yes

Reviewer #2: Yes

3. Has the statistical analysis been performed appropriately and rigorously?

Reviewer #1: Yes

Reviewer #2: Yes

4. Have the authors made all data underlying the findings in their manuscript fully available?

Reviewer #1: Yes

Reviewer #2: Yes

5. Is the manuscript presented in an intelligible fashion and written in standard English?

Reviewer #1: Yes

Reviewer #2: Yes

Reviewer #1: (No Response)

Reviewer #2: The manuscript is acceptable for publication in the present form. The authors have answered all questions/comments.

**Do you want your identity to be public for this peer review?** For information about this choice, including consent withdrawal, please see our Privacy Policy

Reviewer #1: **Yes: ** Sue Faye Siow

Reviewer #2: No

---

## [Editor Report · Acceptance letter]

PONE-D-25-06200R1

PLOS ONE

Dear Dr. Saute,

I'm pleased to inform you that your manuscript has been deemed suitable for publication in PLOS ONE. Congratulations! Your manuscript is now being handed over to our production team.

Kind regards,

on behalf of

Dr Jian Jing

Academic Editor

PLOS ONE